# Quality of Life of Patients with Glaucoma in Slovakia

**DOI:** 10.3390/ijerph18020485

**Published:** 2021-01-09

**Authors:** Ľudmila Majerníková, Anna Hudáková, Andrea Obročníková, Beáta Grešš Halász, Mária Kaščáková

**Affiliations:** Department of Nursing, Faculty of Health Care, University of Prešov, Partizánska 1, 08001 Prešov, Slovakia; andrea.obrocnikova@unipo.sk (A.O.); beata.gress.halasz@unipo.sk (B.G.H.); maria.kascakova@unipo.sk (M.K.)

**Keywords:** primary glaucoma with open iris-corneal angle, visus, perimeter, quality of life, WHOOQL-BREF, NEI VFQ-25

## Abstract

*Purpose:* The aim of this study was to identify and analyse the quality of life of patients with primary open-angle glaucoma (POAG) based on their visus and peripheral vision. *Methods:* Our study was observational in nature; it was a cross-sectional study. In total, 119 patients with POAG were included in a causal-comparative character, ex post facto research design. The authors collected data using the National Eye Institute Visual Function Questionnaire-25 (NEI VFQ-25) and World Health Organization Quality of Life abbreviated version questionnaire (WHOQOL-BREF) tools. *Results:* Only patients with POAG that were over 18 years of age and had no other ocular or chronic illnesses were included. The mean duration of glaucoma was 8.77 (SD ± 5.63) years. Binocular disability was observed in 68.0% of patients. Using WHOQOL-BREF, there were significant differences found in the better-eye-vision group in psychological (*p* < 0.001) and environment (*p* < 0.001) domains. In the worse-eye-vision group, significant differences were found in physical health (*p* < 0.001), environment (*p* < 0.001), and quality related to health (*p* < 0.001) domains. Using NEI VFQ-25, there were significant differences found (*p* = 0.000) in all domains except subscale driving. *Conclusion:* Quality of life of patients with visual impairment is significantly lower in comparison to that of patients without a visual impairment.

## 1. Introduction

In the health sciences, the assessment of quality of life is currently popular, particularly in the field of nursing research. The quality of life of a patient with visual impairment can be assessed from several aspects. Vision-protection methods act to preserve vision from losses caused by glaucoma. The main objective of treatment is to maintain patients’ visual functions while considering the quality of their lives. The major threat to patients with visual impairment is a gradual loss of vision. A diagnosis of a chronic and potentially blinding disease causes anxiety and fear and requires some degree of acceptance and adaptation to changed visual perceptions. Changes in the mental state of patients are often related to the fear of progression of the disease and the possible permanent loss of vision. If visual impairment occurs, anger over this impairment can have a negative impact on the quality of life associated with vision as well as on general quality of life [1]. From a functional point of view, in the early stages, glaucoma affects peripheral vision and in later stages, affects central visual acuity. Other functional visual impairments may include defects of colour perception, contrast sensitivity, and adaptation in darkness. These altered visual functions have an impact on orientation in the field, activities of everyday life (in relation to the near and distant vision), and colour vision. Sensory deprivation is also related to a lower quality of life. Changes in self-sufficiency, self-reliance, social and work enforcement, social status, and prosperity are other aspects of the quality of life of a visually impaired patient. Early diagnostics and subsequent accurate treatment are the only methods of preventing vision damage. Despite advancements in glaucoma treatment, the global burden for society remains high and is expected to increase. Blindness and visual impairment are significantly associated with higher medical care costs, more days of informal care, and decreased health fitness. Frick, Gower et al. conducted a survey related to medical expenditures to obtain an estimate of the relationships among visual impairment and the blindness and total medical expenses, components of expenses, informal care days, and health benefits. Home care and treatment costs were associated to the largest extent with loss of vision. The total annual economic impact included $5.5 billion spent on medical care and on informal care in the US [2]. Primary open angle glaucoma (POAG) is the most common type of glaucoma, representing three quarters (74%) of all glaucoma cases. The prevalence of POAG is highest in Africa, and the prevalence of primary angle-closure glaucoma (PACG) is highest in Asia.

In 2013, an estimated 64.3 million people aged 40–80 years were affected by glaucoma; the authors of [3] estimated that by 2020, 79.6 million people would be affected, and by 2040, 111.8 million people would be affected [3]. In 2009, in Slovakia, there were 144,292 patients with glaucoma and 16,506 of those were newly diagnosed cases. For each 100,000 inhabitants of Slovakia, 3352 people in 2009 were affected by glaucoma [4]. Because of the population development and demographic indicators, the incidence of glaucoma in Slovakia is increasing. In 2016, the number of individuals with glaucoma was almost double that of 2009, with 235,060 total cases and 33,515 newly reported cases. As of 2016, there were 4325 registered patients with glaucoma per 100,000 inhabitants in Slovakia [5].

### 1.1. Materials and Methods

After obtaining permission for the research from the Ethics Committee of The Autonomous Region of Prešov, we conducted the research from June to August 2018. The protocol number assigned by the Ethics Committee (EC Slovakia) was No: 483/2018/OZ-058. We received approval for our research.

The tenets of the declaration of Helsinki were adhered to while conducting this research. We approached selected private ophthalmology out-patient clinics and asked the professional guarantors for consent to conduct the research. Two standardized questionnaires were distributed within the research process: NEI VFQ-25 (National Eye Institute Visual Function Questionnaire-25) and WHOOQL-BREF (World Health Organization Quality of Life abbreviated version questionnaire). Patients were recruited according to predefined criteria (age, diagnosis, informed consent to participate in this research).

Patients with severe comorbidities that could significantly affect the outcome of the visual function questionnaire, such as serious mental and systemic diseases, were not included in the study. This study included 119 patients with POAG. The response rate was 100.0%, as questionnaires were completed by means of an individually-managed interview with each patient in person during the period of June to August 2018.

Each patient was informed of the purpose of the data collection, their anonymity, and how the results of the research project would be used. Each patient signed the informed consent to participate in this research and to disclose additional data from his/her health-care documentation. The directed interview with each respondent lasted 20 to 40 min.

Reliability coefficients (Cronbach’s alpha) for individual scales ranged from 0.62 to 0.78.

### 1.2. Characteristics of Used Methodology

To assess the quality of life, we used two standardized questionnaires. For better understanding, faster responses, and to assist individuals with possible visual impairments, the questionnaires were completed with the assistance of an interviewer.

#### 1.2.1. NEI VFQ-25 (Slovak Version)

The Slovak version of the NEI-VFQ 25 was translated from the original English version. Our NEI VFQ-25 questionnaire is an abbreviated version of the original 51-item questionnaire. NEI VFQ-25 is a validated questionnaire and measurement tool for the quality-of-life assessment related to visual functions. It was developed and put into practice in its English version by RAND in cooperation with the NEI (National Eye Institute) in the United States of America. It has now been validated for use in multiple languages and has become a part of many studies. Our translated questionnaire itself as well as the permission for its use for our research project were obtained from MUDr. Erika Vodrážková, MPH. Two independent accredited translators carried out the translation. Taking into account the translators´ suggestions, we altered the outdoor activities in the Slovak version of question A7 to walking, cycling, tourism, work in the garden, as opposed to the activities bowling, running, golf [6].

#### 1.2.2. WHOQOL-BREF (Slovak Version)

We created a shortened version of the WHOQOL-BREF questionnaire for clinical practice that is currently available in 50 language versions, including the Czech language. It consists of four domains containing 24 questions and 2 separate questions focused on the assessment of quality of life and health satisfaction. We used the validated version of the questionnaire for the purposes of our work with consent of the authors Dragomerická and Bartoňová from 2006 [7].

Despite the division into two groups, an 80% statistical force was retained to capture even small effects.

### 1.3. Characteristics of Statistical Procedures

For the analysis of sociodemographic indicators and other characteristics of the research sample, we used descriptive statistics (frequency, percentage, mean, and standard deviation). To detect the existence of significant differences among independent groups, we used the Mann–Whitney U and Kruskal–Wallis tests. The aim was to find out whether the differences among medians of each group were statistically significant within the sample, suggesting the existence of a relationship among variables (*p* < 0.05 * significant, *p* < 0.01 ** highly significant, *p* < 0.001 *** very highly significant, *p* < 0.0001 *** very, very highly significant). Reliability was determined by Cronbach alpha coefficient and reached the value of 0.72 (sufficient internal consistency of the scale) [8].

## 2. Results

Our study sample consisted of 119 patients with primary open-angle glaucoma (POAG). The study included patients diagnosed with and treated for POAG who did not suffer from other ocular illnesses, other chronic illnesses, were at least 18 years of age, and were outpatients of ophthalmologists who agreed to participate in the research. Research has a causal-comparative character and ex post facto method.

### 2.1. Sociodemographic Data

Sociodemographic data provides a comprehensive picture of the research sample. We collected general demographic data including gender, age, and education of respondents. We collected specific demographic data including visus and peripheral vision, length of treatment, and POAG diagnosis depending on the impairment of the eye. A detailed analysis of the demographic data is provided by the following descriptive evaluation (Table 1). Female respondents (65.0%) predominated in our sample. The majority of respondents were secondary education graduates with a maturity exam (28.0%), followed by those with a university education (21.0%) and those with a primary education (18.0%). The mean age was 59.4 (SD ± 18.32) years, with the majority being older than 70 years. The mean duration of glaucoma was 8.77 (SD ± 5.63) years, with most respondents having the disease for less than 5 years (30.0%). We observed binocular disability in 68.0% of respondents with glaucoma (Table 1).

The visual acuity (visus), as well as the range of the peripheral vision, are important diagnostic criteria for glaucoma. We assessed the status of the visual function of the respondents by objective measurements: examination of the visual acuity/visus (Snellen optotypes) and assessment of peripheral vision status (computer perimeter). For the visus examination results, we categorised our respondents with visual impairment based on the World Health Organization (WHO) Classification of Vision Impairment (Table 2 and Table 3).

The comparison was based on the results of the visus between the eye with better vision (EBV) and the eye with worse vision (EWV). We found that the EWV was not impaired in 97 cases and pathological findings were observed in 22 cases. We detected the EBV was not impaired in 100 cases and had pathological findings in 19 cases (Table 2). Based on the peripheral vision results, we noticed a loss of field of view in the EWV (*n* = 67) and EBV (*n* = 67). Pathological changes noticed in the peripheral vision were found in the EWV (*n* = 52) and EBV (*n* = 52) (Table 3). Respondents with glaucoma were divided into two groups based on the level of visual acuity: patients with physiological visus and patients with pathological findings that could not be sufficiently corrected. Using the WHOQOL-BREF questionnaire, we researched and compared responses of these two groups in terms of significantly different responses within the measured domains of quality of life. In Table 4, we present significant differences using the Mann–Witney U test in perceived quality of life in the following domains: D1 domain, physical health (EBV *p* < 0.001; EWV *p* < 0.01); D2 domain, psychological health (EBV *p* < 0.001; EWV *p* < 0.01); D4 domain, environment (EBV *p* < 0.001; EWV *p* < 0.001); Q1, quality of life (EBV *p* < 0.01; EWV *p* < 0.01); and Q2, health (EBV *p* < 0.01; EWV *p* < 0.001). These results were found in relation to pathological changes recognised in the eye with better vision and worse vision in terms of physiological/pathological visus. The results point to a higher level of respondents´ perception of quality of life in those who had well-compensated eyesight. Only in the D3 domain, social relations, were differences not confirmed. In terms of quality of life of patients with glaucoma and their perception assessment, domains D4, environment (M = 16.92); D3, social relations (M = 16.12); and D2, psychological health (M = 16.01) were evaluated as the best from the aspect of the EWV patients with physiologically-compensated vision. Respondents with a pathological visus rated the best domains as D3, social relations (M = 15.73); D4, environment (M = 16.92); and D2, psychological health (M = 14.52). Both groups identified domain D1, physical health, as the worst field of quality of life, where the mean value of the responses of the patients with physiological visus was M = 14.28 and that of patients with pathological visus was M = 12.0 (Table 5). According to the assessment of quality-of-life perception of patients with glaucoma in this study, patients with physiologically compensated visus rated domains D4, environment (M = 16.98); D2, psychological health (M = 16.12); and D3, social relations (M = 15.94) the best. Respondents with pathological visus rated domains D3, social relations (M = 15.65); D4, environment (M = 14.49); and D2, psychological health (M = 14.33) the best. As the worst field of quality of life, both groups identified domain D1, physical health, with the mean value of responses of the patients with physiological visus M = 15.12 and that of patients with pathological visus M = 12.07.

### 2.2. WHOQOL-BREF Questionnaire Evaluation—Peripheral Vision

The level of damage based on the results of peripheral vision of the EBV and EWV was classified as 0: no loss of field of visus, 1: moderate changes, 2: advanced changes. In these groups, we compared the quality of life using WHOQOL-BREF, assuming a better quality of life for respondents with physiological findings or slightly advanced changes in their visus. In Table 6 we present statistically significant differences in the quality-of-life perception. Respondents were divided into groups on the bases of peripheral vision. The analysis was conducted using a Kruskal–Wallis test. The results show significant differences in the perception of quality of life in all domains. Patients with lower pathogenicity expressed better ratings of each field of quality of life than did respondents with more severe vision impairment. In EBV patients, we noticed significant differences (Table 4) at *p* < 0.001 in domains D1, physical health; D4, environment; Q1, quality of life; and Q2, health, at level *p* < 0.01 in domain D2, psychological health, and at level *p* < 0.05 in domain D3, social relations.

In EWV patients (Table 5) we noticed significant differences at *p* < 0.001 in domains Q1, quality of life and Q2, health, at level *p* < 0.01 in domains D1, physical health; D2, psychological health; D4, environment, and at level *p* < 0.05 in domain D3, social relations.

### 2.3. NEI VFQ-25 Questionnaire Evaluation—Peripheral Vision

In Table 6 and Table 7 we present an analysis of statistically significant differences in perception of the quality of life represented among groups of respondents based on peripheral vision using NEI VFQ-25. The results show significant differences in the perception of quality of life in various domains. When comparing three groups, we noticed differences in the perception of the quality of life in all subscales (from the perspective of pathology/physiology) on the right as well as the left eye except for the subscale 10, driving. Patients with lower pathogenicity expressed better rating in each subscale of quality of life compared to respondents with advanced visual changes. In all our subscales, we discovered significant differences in the quality-of-life domains evaluated at level *p* < 0.001.

## 3. Discussion

The importance of this work lies in the fact that the quality of life of patients with glaucoma has not been investigated so far in the Slovak population. This issue was dealt with by authors in the Czech Republic [9], who completed a pilot study involving only 20 patients with glaucoma. However, there are many studies in the literature from other countries that actively deal with this issue [10,11,12,13,14,15,16,17,18,19]. Patients with binocular loss of visual functions have serious difficulties with daily activities such as reading, motion and orientation in the field, and driving [20,21,22]. However, the quality of life (QoL) can also be affected by monocular loss of visual functions. In our study, we found statistically significant differences in the respondents’ perception of the quality of life in individual domains. Significant differences comparing visus of the EBV and EWV were discovered in all domains except domain D3, social relations. The greatest significant differences were in domain D4, environment, in relation to EBV, and in domain D1, physical health, in relation to EWV. Many foreign authors have assessed the quality of life of patients with POAG. Grow et al. in their study examined respondents with deteriorated socio-economic status [23]. The study involved 190 patients with different eye diseases in the age group of 60–107 years. Of the total number of respondents, 38.0% were POAG patients, and 41% of them had visual impairment problems, specifically of peripheral vision. The results of the study showed no significant statistical differences between the two groups depending on age, gender, marital status, income, or perceived economic need. We identified significantly different quality-of-life indicators in patients with POAG. In our sample, statistical significance was found in almost all domains. The above-mentioned authors recorded this statistical significance in domains of physical and psychological health only. For comparison, a study by Kumari et al. reviewed patients divided into groups by types of disease. The study included 50 POAG patients and 50 patients with cataract disease as a control group. Patients with POAG reported statistically significant differences in all domains, including overall health status and quality of life differences in comparison to the group of patients with cataract disease (*p* < 0.05) [24]. Because of its asymptomatic chronic nature and the potential loss of vision, glaucoma is a psychological burden [25,26]. Restrictions on living caused by various factors, such as driving restrictions [22,27], fear of falls [28,29], and worsening of balance [30], also contribute to the relationship between glaucoma and depression. Authors Berdeaux et al. included 60 patients each of the following groups: patients with POAG, patients with age-related macular degeneration (AMD), and a control group of patients assessed for eye correction. The aim of the study was to identify signs of depression, anxiety, and quality of life in POAG patients compared to that in patients with age-related macular degeneration (AMD). The symptoms of depression and anxiety were evaluated using the Hamilton Depression Rating Scale (HDRS) and the Hamilton Anxiety Rating Scale (HARS). The analysis of the results revealed that scores of patients with POAG and AMD in the domains of physical health, social relationships, the environment, and psychological health were significantly lower than those in the control group (*p* < 0.05). Significant differences in the POAG group and in the AMD group (*p* < 0.05) were not confirmed in any domains. A notifiable reduction in the quality of life of POAG and AMD patients was found. They were more depressed and anxious. From the authors’ point of view, it is necessary to analyse the current psychological state of patients as an important predictor of QoL in terms of prognosis. A specialist must continuously monitor patients with psychical symptoms and ophthalmic diseases (glaucoma, AMD) [31]. The highest level of quality of life within the whole sample of respondents was noticed using the NEI VFQ-25, specifically in the subscales 11, colour vision; 6, social functioning; and 10, driving. Toprak et al. also examined the overall assessment of visual functions in their research study. The authors found the highest scores in subscales 11, colour vision (100) and 5, distance vision (94.4). In contrast, the lowest score was observed in subscale 3, ocular pain (57.0) [16]. For comparison, we analysed the results of a pilot study from the Czech Republic by Skorkovská et al. The examined sample consisted of 20 patients with POAG. The mean age of the subjects in the sample was 70.05 years (45–87). The lowest scores we obtained in the subscales 1, health in general (48.3); 9, dependency (58.3); 2, vision in general (61.4); and 7, mental health (61.4). In contrast, subscales that remained almost unchanged were 11, colour vision (100); 6, social functioning (95.3); and 10, driving (94.4) [9]. In a further study by Floriani et al., the authors evaluated the results obtained on a sample of 3169 patients with POAG using the NEI VFQ-25 questionnaire. The mean age of the patients was 66.9 years. In their study, the authors highlighted the high quality of life scores at the onset of the disease, while the progressive severity of the disease caused a decline in quality of life. The worst score we observed in the subscale vision in general (69.2). Patients reported markedly higher scores in all domains with the disease in stage 0 (newly diagnosed glaucoma without damage to the optic nerve structures) versus stage 5 (severe damage to the optic and peripheral vision structures). The authors point out the need of early diagnostics and appropriate and effective treatment [12]. From the above results, it is clear that patients with glaucoma generally perceive health as a negative state that causes psychic reactions (anger, anxiety, and hostility). The pathological eye’s condition corresponds with the definition of the WHO from the perspective of optimal health status as “the state of complete physical, mental and social well-being and not only the absence of disease or disability” [9]. In our results recorded by the NEI VFQ-25 questionnaire, we confirmed statistically significant differences in the perception of quality of life based on the sample division from the point of view of peripheral vision pathology to EBV and EWV. In comparison of three groups, we noticed differences in the perception of quality of life in all subscales, except the subscale driving. Patients with lower pathogenicity expressed better results in each subscale of quality of life than did respondents with more severe vision impairment. Similarly, Wolfram et al. conducted a similar study. The authors divided the sample of patients with POAG into three groups based on peripheral vision changes: low, moderate, and advanced impairment. Data indicated a low score for domain 1, health in general (low 60.1, moderate 52.3, and advanced 44.4). Similarly, low values were noticed in the assessment of domain 2, vision in general (low 83.8, moderate 76.9, and advanced 65.3) and domain 3, ocular pain (low 87.2, moderate 81.3, and advanced 75.8). The highest rating in other subscales was represented by patients with POAG at low (100–87.2) and moderate impairment (91–88). The advanced POAG patients (60–63) presented the lowest values [32]. The results show that the progression of the disease has an adverse effect on the quality of life. Their results correspond not only to our results but also to the results of other authors [9,12,16]. Finger et al. conducted a cross-sectional study on a sample of 1085 patients with various ocular diseases including glaucoma, of which 254 were without ocular pathology (*n* = 543 in Australia and *n* = 796 in Germany). Differences in QoL assessed by the VisQoL instrument were statistically significant in relation to an eye with better vision compared to an eye with worse vision with varying degrees of vision impairment. The generic EQ-5D instrument did not confirm the changes among the two group within the visual field. QoL indicators were confirmed in patients with diabetes (*p* < 0.05), while the QoL score was significant in terms of the variables sex, age (*p* < 0.001), and visual acuity (*p* < 0.001). Results based on visual acuity using a generic tool are likely to underestimate the effect of visual impairment, especially if a better eye has no or minimal loss of visual acuity and the worse eye is mildly-to-severely visually impaired [33]. Sawada et al. focused on two groups of patients: those with normotensive glaucoma (*n* = 84) and those POAG (*n* = 84). The mean age of patients was 61.5 years. The quality of life was evaluated based on the results of the visus and the peripheral vision, dividing the patients into EBV patients and patients with EWV. Data collection was performed based on corrected visual acuity, which was measured as a logarithm of the minimum resolution angle (log10MAR) in both eyes in all patients. The authors’ statement confirms the highly significant relationship between the quality of life and the results of the visus and the peripheral vision. With EBV based on the visus, the authors found statistically significant differences in four subscales: 2, vision in general; 5, distance vision; 4, near vision; and 10, driving (*p* < 0.001). In the case of EWV, the authors found a very high significance (*p* < 0.001) in 5, distance vision and 4, near vision, and statistically high significance also in the evaluation of 2, vision in general (*p* < 0.01). The authors in the assessment of peripheral vision reported similar results [18]. Gillespie et al. in their American study evaluated 401 patients with POAG in terms of quality of life by the NEI VFQ-25. The mean age of respondents was 58.0 years. The results of their study showed that, based on EBV assessment, statistical significance was not confirmed in subscale 11, colour vision, but the statistical significance was confirmed in all other subscales. The highest scores in the results were reported by the authors in the overall health assessment, in the assessment of the difficulties in the individual roles, in driving, in mental health, in peripheral vision, in near vision, in distant vision, and in social functions. These results were statistically very significant (*p* < 0.001) [13].

## 4. Discussion

From the analysis of the results of this study, we confirmed that patients with impaired vision subjectively evaluated their quality of life more negatively than those with better vision. The results were very significant in all dimensions of quality of life using the WHOQOL-BREF and NEI VFQ-25 instruments. We declare the need to promote active screening of glaucoma in the Slovak Republic and to examine the issue of quality of life of patients with eye diseases, including glaucoma, in the Slovak population. Christen et al. examined the impact of multivitamin use on cataract incidence and age-related macular degeneration (AMD). Over 11.2 years of research on the health status of men aged less than 50 years of age receiving multivitamins, and men taking placebo, there was a significant difference in the reduction of cataract and AMD risk found, particularly between the group of respondents taking multivitamins including 872 patients with cataracts, and 945 placebo-treated cataract patients (*p* = 0.04), and in the case of AMD, 152 patients receiving multivitamins and 129 cases receiving placebo (*p* = 0.15) [34]. This study provided interesting results. Hypothetically, it would be appropriate to carry out a similar study in patients with glaucoma focusing on prevention.

## Figures and Tables

**Table 1 ijerph-18-00485-t001:** Sociodemographic data.

Demographic Data	*n*	%
Sex		
Male	42	35
Female	77	65
Education		
Primary	21	18
Vocational without maturity exam	16	13
Vocational with maturity exam	13	11
Secondary vocational without maturity exam	10	8
Secondary vocational with maturity exam	34	28
University	25	21
Age (M ± SD)	59.4 SD ± 18.32 years	
Less than 50 years	27	23
51–59	22	18
60–69	33	28
More than 70 years	37	31
Duration of disease (M ± SD)	8.77 SD ± 5.63 years	
One year	20	17
Less than 5 years	36	30
Less than 10 years	33	28
11 years or more	30	25
Diagnosed		
Both eyes	79	68
Right eye	26	22
Left eye	11	9

**Table 2 ijerph-18-00485-t002:** Frequency of respondents within visual impairment (visus) categories. EWV: eye with worse vision; EBV: eye with better vision.

Classification of Vision Impairment (Visus)	EWV *n*	EBV *n*
0 without visual impairment (normal vision)	97	100
1 6/18–6/60 near-normal vision	14	9
2 6/60–3/60 moderate vision impairment	2	4
3 3/60–1/60 severe vision impairment	3	3
4 1/60, 1/50 moderate blindness	3	3
5 severe to total blindness	0	0
Total	119	119

**Table 3 ijerph-18-00485-t003:** Frequency of respondents within visual impairment (peripheral vision) categories.

Classification of Vision Impairment (Peripheral Vision)	EWV	EBV
0 without visual impairment	67	67
1 slightly advanced changes, mild loss of field of view (40–30° from the centre of fixation)	24	25
2 advanced changes, moderate loss of field of view (20–10° from the centre of fixation)	28	27
3 concentric narrowing of the field of view (15–5° from the centre of fixation)	0	0
4 remnants of the field of view in absolute glaucoma (less than 5° from the centre of fixation)	0	0
Total	119	119

**Table 4 ijerph-18-00485-t004:** Differences in the domains of the researched groups in terms of physiological/pathological visus.

Rating on EBV					Rating on EWV			
	Z	*p*	n VPa	n VPh	Z	*p*	n VPa	n VPh
D1	−3.268	0.00108 **	19	100	−3.638	0.0002 ***	22	97
D2	−3.612	0.00030 ***	19	100	−2.762	0.0057 **	22	97
D3	−1.599	0.1096	19	100	−1.516	0.1294	22	97
D4	−3.348	0.00081 ***	19	100	−3.344	0.00082 ***	22	97
Q1	−3.09	0.0019 **	19	100	−2.738	0.0061 **	22	97
Q2	−3.192	0.0014 **	19	100	−3.556	0.0003 ***	22	97

Notes: Z-coefficient; ** *p* < 0.01; *** *p* < 0.001; VPa visus pathologic; VPh—visus physiologic. D1—physical health; D2—psychological health; D3—social relations; D4— environment; Q1—domain quality of life; Q2—domain health.

**Table 5 ijerph-18-00485-t005:** Differences in the means of the domains of the researched groups in terms of physiological/pathological visus.

**Rating on EBV**	**Pathological Visus**			**Physiological Visus**		
	**M**	**SD**	**Min–Max**	**M**	**SD**	**Min–Max**
D1	12.07	1.86	8–16.35	15.12	1.94	9.45–16.94
D2	14.33	1.72	9.65–19.33	16.12	2.02	9.01–20
D3	15.65	1.97	8.5–20	15.94	2.55	9.67–20
D4	14.49	2.35	9.48–18.79	16.98	2.62	8.82–20
Q1	4.00	0.78	1–5	4.52	0.89	2–5
Q2	2.59	0.89	4–2	4.0	0.88	1–5
**Rating on EWV**	**Pathological Visus**			**Physiological Visus**		
	**M**	**SD**	**Min–Max**	**M**	**SD**	**Min–Max**
D1	12.0	1.82	8–16.35	14.28	1.95	9.98–17.82
D2	14.52	1.78	9.28–19.24	16.01	1.99	9.01–20
D3	15.73	2.01	8.1–20	16.12	2.52	10.1–20
D4	14.59	2.25	9.48–18.79	16.92	2.72	8.79–20
Q1	4.01	0.88	1–5	4.5	0.79	2–5
Q2	2.59	0.96	4–2	4.0	0.96	1–5

Notes: M—mean; SD—standard deviation; Min–Max—minimum and maximum value. D1—physical health; D2—psychological health; D3—social relations; D4—environment; Q1—domain quality of life; Q2—domain health.

**Table 6 ijerph-18-00485-t006:** Statistical evaluation of WHOQOL-BREF by Kruskal–Wallis test of the EBV and EWV: peripheral vision.

**EBV** **Peripheral Vision**					
**Rating in Fields**	**Perimeter°**	**Mean Rank**	**Kruskal–Wallis**	**Df**	***p***
D1	0	71.1	19.31	3	0.0001 ***
	1	51.2			
	2	37.5			
D2	0	69.4	12.56	3	0.0019 **
	1	49.1			
	2	44.8			
D3	0	67.2	9.91	3	0.049 *
	1	51.2			
	2	47.0			
D4	0	70.4	15.19	3	0.0005 ***
	1	46.5			
	2	44.5			
Q1	0	69.5	15.40	3	0.0005 ***
	1	49.3			
	2	44.4			
Q2	0	71.7	22.32	3	0.000 ***
	1	43.0			
	2	44.9			
**EWV** **Peripheral Vision**					
**Rating in Fields**	**Perimeter°**	**Mean Rank**	**Kruskal–Wallis**	**Df**	***p***
D1	0	68.6	10.38	3	0.0056 **
	1	55.6			
	2	44.8			
D2	0	69.7	12.03	3	0.0024 **
	1	51.1			
	2	44.8			
D3	0	68.6	9.91	3	0.043 *
	1	53.5			
	2	46.4			
D4	0	68.6	10.05	3	0.007 **
	1	53.5			
	2	51.0			
Q1	0	70.7	16.44	3	0.0003 ***
	1	47.3			
	2	47.0			
Q2	0	71.9	20.42	3	0.000 ***
	1	43.8			
	2	47.9			

Notes: * *p* < 0.05; ** *p* < 0.01; *** *p* < 0.001. D1—physical health; D2—psychological health; D3—social relations; D4—environment; Q1—domain quality of life; Q2—domain health.

**Table 7 ijerph-18-00485-t007:** Statistical evaluation of the NEI VFQ-25 by Kruskal–Wallis test of the EBV and EWV: peripheral vision.

**EBV** **Peripheral Vision**					
**Rating in Fields**	**Perimeter°**	**Mean Rank**	**Kruskal–Wallis**	**Df**	***p***
Subscale 1	0	68.56	12.10	3	0.0024 **
	1	52.19			
	2	43.83			
Subscale 2	0	67.75	26.00	3	0.000 ***
	1	66.46			
	2	30.70			
Subscale 3	0	68.53	20.67	3	0.000 ***
	1	60.07			
	2	35.37			
Subscale 4	0	70.30	37.41	3	0.000 ***
	1	62.13			
	2	28.06			
Subscale 5	0	73.89	43.61	3	0.000 ***
	1	54.59			
	2	25.91			
Subscale 6	0	70.2	43.22	3	0.000 ***
	1	62.0			
	2	36.4			
Subscale 7	0	74.55	40.92	3	0.000 ***
	1	54.21			
	2	24.43			
Subscale 8	0	72.18	50.00	3	0.000 ***
	1	59.76			
	2	25.22			
Subscale 9	0	69.21	63.50	3	0.000 ***
	1	67.42			
	2	25.47			
Subscale 10	0	26.69	2.46	3	0.292
	1	20.66			
	2	37.5			
Subscale 11	0	68.68	51.03	3	0.000 ***
	1	63.04			
	2	31.72			
Subscale 12	0	72.02	52.78	3	0.000 ***
	1	60.15			
	2	25.29			
**EWV** **Peripheral Vision**					
**Rating in Fields**	**Perimeter°**	**Mean Rank**	**Kruskal–Wallis**	**Df**	***p***
Subscale 1	0	69.8	16.69	3	0.000 ***
	1	57.7			
	2	40.5			
Subscale 2	0	70.7	30.47	3	0.000 ***
	1	70.2			
	2	32.5			
Subscale 3	0	70.5	24.01	3	0.000 ***
	1	59.9			
	2	37.2			
Subscale 4	0	71.2	24.01	3	0.000 ***
	1	64.6			
	2	37.2			
Subscale 5	0	77.8	56.96	3	0.000 ***
	1	53.3			
	2	26.7			
Subscale 6	0	70.2	43.22	3	0.000 ***
	1	62.0			
	2	36.4			
Subscale 7	0	75.7	46.07	3	0.000 ***
	1	60.5			
	2	25.6			
Subscale 8	0	73.4	48.60	3	0.000 ***
	1	60.7			
	2	30.1			
Subscale 9	0	70.67	50.47	3	0.000 ***
	1	64.33			
	2	33.40			
Subscale 10	0	27.7	2.71	3	0.235
	1	18.08			
	2	25.33			
Subscale 11	0	68.62	37.04	3	0.000 ***
	1	62.5			
	2	38.25			
Subscale 12	0	74.91	58.57	3	0.000 ***
	1	58.20			
	2	29.11			

Notes: ** *p* < 0.01; *** *p* < 0.001.

## Data Availability

The data presented in this study are available on request from the corresponding author. The data are not publicly available due to General Data Protection Regulation (GDPR).

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
