# Peer review of "Quality of Life of Patients with Glaucoma in Slovakia"

_ijerph, 2021, doi:10.3390/ijerph18020485_

Round 1
Reviewer 1 Report
Dear authors,
I attach the recommendations and considerations about your paper.
Best regards.

Author Response
Dear reviewer
Thank you for your comments. I corrected the article according to your recommendations.
Yours sincerely
Anna Hudáková and team of authors

Reviewer 2 Report
COMMENTS TO AUTHORS:
This paper purposed to identify and analyse the quality of life of patients with primary open-angle glaucoma (POAG) based on their visus and perimeter. They also found that patients with impaired vision have subjectively evaluated their quality of life more negatively than those with better vision. I do have some comments as listed below in the order noted.
Comment 1:
The quality of the data set is very important, especially in the selected private ophthalmology out- patient clinics. For this reason, please clarify the included criteria and excluded criteria of sample collection in the section of Materials and Methods and please provide a flowchart immediately at the end of the section.
Comment 2:
Sample size calculation is very important. It is not described how sample calculation was performed and the statistical power in this study.
Author Response
Dear Reviewer
Thank you for your commnets.
I corrected the article according to your recommendations.
Yours sincerely
Anna Hudáková and team of authors
